# Analysis of Sequential Micromixing Driven by Sinusoidally Shaped Induced-Charge Electroosmotic Flow

**DOI:** 10.3390/mi13111985

**Published:** 2022-11-16

**Authors:** Haizhen Sun, Ziyi Li, Yongji Wu, Xinjian Fan, Minglu Zhu, Tao Chen, Lining Sun

**Affiliations:** 1School of Mechanical and Electric Engineering, Soochow University, Suzhou 215299, China; 2Jiangsu Provincial Key Laboratory of Advanced Robotics, School of Mechanical and Electric Engineering, Soochow University, Suzhou 215123, China; 3School of Future Science and Engineering, Soochow University, Suzhou 215299, China

**Keywords:** multiple parallel streams, induced charge electroosmosis, microvortex development, on-demand sequential mixing

## Abstract

Multi-fluid micromixing, which has rarely been explored, typically represents a highly sought-after technique in on-chip biochemical and biomedical assays. Herein, we propose a novel micromixing approach utilizing induced-charge electroosmosis (ICEO) to implement multicomplex mixing between parallel streams. The variations of ICEO microvortices above a sinusoidally shaped floating electrode (SSFE) are first investigated to better understand the microvortex development and the resultant mixing process within a confined channel. On this basis, a mathematical model of the vortex index is newly developed to predict the mixing degree along the microchannel. The negative exponential distribution obtained between the vortex index and mixing index demonstrates an efficient model to describe the mixing performance without solving the coupled diffusion and momentum equations. Specifically, sufficient mixing with a mixing index higher than 0.9 can be achieved when the vortex index exceeds 51, and the mixing efficiency reaches a plateau at an AC frequency close to 100 Hz. Further, a rectangle floating electrode (RFE) is deposited before SSFE to enhance the controlled sequence for three-fluid mixing. One side fluid can fully mix with the middle fluid with a mixing index of 0.623 above RFE in the first mixing stage and achieve entire-channel mixing with a mixing index of 0.983 above SSFE in the second mixing stage, thereby enabling on-demand sequential mixing. As a proof of concept, this work can provide a robust alternative technique for multi-objective issues and structural design related to mixers.

## 1. Introduction

In the past decade, “lab on a chip” (LOC) technology has developed in a robust fashion to facilitate broad applications in chemical, biological, and material assays [1,2,3]. Considering the sample consumption, analysis time, and device integration, LOC has many advantages over conventional systems. Of special interest is micromixing, which aims to make the molecules from different reagents contact rapidly within a microchannel [4,5]. In particular, multiple micromixing is crucial for multicomponent reactions [6], organic synthesis [7], and detection of highly complex biomacromolecules [8]. Up until now, most studies on micromixing have mainly focused on two-fluid mixing within a single channel [9,10] or segmented-channel mixing for multiple fluids [11,12]. However, multi-fluid micromixing, especially for the sequential micromixing of parallel flows, has rarely been studied. Therefore, there is still a need for techniques that can flexibly implement continuous multi-fluid micromixing. 

Generally, conventional sequential mixing requires multi-step mechanical agitation, bearing the risk of reagent consumption and inhomogeneous concentration [13]. Especially for a fast reaction or quantitative detection, the reactant amount cannot be precisely controlled due to the insufficient mixing process. To circumvent the limitations of conventional mixing, fluidic mixing within a microscale channel, both passive and active, has attracted tremendous attention and has been demonstrated to efficiently enhance mass transfer [14,15]. Specifically, passive micromixing usually relies on the special geometries embedded in a microchannel or complex structures of the microchannel [16,17]. In addition to the complicated fabrication process, such an approach is typically used to implement multi-step or simultaneous micromixing between three or more flow streams [18,19]. Sequential micromixing between multiple parallel streams is still challenging. By contrast, active micromixing is driven by external energy sources, such as magnetic fields [20,21], acoustic energy [22,23], electric stimulation [24,25,26,27], and optical fields [28]. An active mixer enables fast and homogeneous micromixing within a simple channel and in a short time. More importantly, it is convenient to control the mixing position and mixing time through the design of external-stimulus implementation. Amongst active micromixers, the electrokinetic approaches, such as induced charge electroosmosis (ICEO) [29,30] and AC electrothermal (ACET) flow [31,32], offer unique features of the simple electrode configuration, no moving parts, and low voltages, making them an attractive mixing mechanism for various bioanalytical applications. Interestingly, the ICEO is generated on an electrically floating electrode that works through the induced voltage instead of directly applying electrical signals, demonstrating that the ICEO-based microvortices could appear anywhere desired. It seems that the ICEO flow can be expected to complete multi-fluid micromixing on demand. To our knowledge, ICEO-based sequential micromixing between multiple streams has not been demonstrated thus far.

In this work, we exploit, for the first time, the reshapable ICEO-based microvortex pair to achieve sequential micromixing. Here, an asymmetrically arranged rectangle floating electrode (RFE) and a subsequent sinusoidally shaped floating electrode (SSFE) within a straight microchannel are designed to induce time-varying microvortex patterns to activate the multi-fluid micromixer. We find that the stagnant line for the microvortex pairs on an SSFE in the flow direction exhibits a sinusoidally shaped morphology and varies with the channel height, proving the capability of transferring chemical species from side to side. To better elucidate the multiple mixing strategy, two-fluid mixing is first investigated. Based on this, a mathematical model of vortex index is proposed to predict the degree of mixing, laying out a new governing rule for accurately quantifying vorticity and micromixing in commonly used vortex-based mixers. Furthermore, three-fluid sequential mixing, accompanied by two contact interfaces disturbed in a controlled sequence, is successfully accomplished. The results demonstrate not only that the middle fluid can mix with side fluids for simultaneous mixing, but also that the side fluids separated by multiple streams can mix with each other step by step. Therefore, the presented multi-fluid mixing strategy can offer a promising LOC alternative for more complex chemical and biological assays.

## 2. Working Principle

The ICEO-based micromixer (Figure 1) comprises two 3D driving electrodes (3DDEs) arranged at both channel sides to generate a strong electrical field across the entire channel, an RFE and an SSFE deposited on the channel bottom to induce time-varying microvortex pattern perpendicular to the flow direction, and a tri-branch microchannel where three streams flow. The RFE is 70 µm wide, 0.75 mm in length, and 185 nm in thickness, and the SSFE is 100 µm wide and 2.5 mm in length. The 3D electrodes are 60 µm in height and 3.4 mm in length. When three fluids with the same conductivity of 10 µS/cm are introduced into the main channel, two contact interfaces are clearly formed. Once an AC signal is applied to the 3DDEs, both RFE and SSFE are polarized and ions in fluids are driven along the electric lines to form a thin double layer on the floating electrode surface. The interfacial jump condition at the double-layer surface is governed by the RC charging equation [33].
(1)$$\sigma n\nabla \tilde{\varphi }=j(2\pi f)C_{0}(\tilde{\varphi }-{\tilde{\varphi }}_{0})$$
where *σ* is the fluid conductivity, *n* is the unit normal vector of the electrode surface, and *f* is the AC frequency. *φ* and *φ*_0_ represent the bulk potential outside the double layer and the induced potential of the floating electrode, respectively. *C*_0_ = *C*_s_*C*_d_/(*C*_s_ + *C*_d_) denotes the equivalent capacitance of the double layer, where *C*_s_ and *C*_d_ are the capacitances of the stern layer and the diffusion layer. 

Outside the double layer, the field lines are expelled and induce two fluidic rolls on the electrode surface. The induced-charge electroosmotic slip is governed by Helmholtz–Smoluchowski formula [34]
(2)$$u_{\mathrm{slip}}=-\frac{\varepsilon_{f}\zeta E_{t}}{\eta }$$
where *ζ* = (*φ*_0_ − *φ*)*C*_s_/(*C*_s_ + *C*_d_) is the induced zeta potential, ***E***_t_ is the tangential electric field component, *ε*_f_ and *η* represent the fluid permittivity and the viscosity, respectively.

In each AC period, the time-averaged ICEO velocity can be derived as [35]
(3)uslip=εfCs2η(Cs+Cd)Re(φ˜−φ˜0)(E˜−E˜⋅n⋅n)*

Considering the steady and incompressible flow, the ICEO flow field can be estimated with the continuity and the momentum equations [36,37]
(4)$$\begin{array}{l} \nabla u=0\\ \rho_{f}(u\cdot \nabla )u-\eta \nabla^{2}u+\nabla p=0 \end{array}$$

Interestingly, the electric field is also significantly affected by the position of the floating electrode, which in turn dominates the formation of the double layer, as shown in Figure 2a. Once the position varies to be asymmetric along the channel width, both the electric field and the double layer are reconstructed. This phenomenon would induce a time-varying microvortex pattern corresponding to the SSFE morphology to alternatively disturb the contact interface, resulting in rapid micromixing between different flow streams (Figure 2b).

In the mixing process, the mass transfer and the molecular concentration can be described by the convection and diffusion equation [38]
(5)$$u\cdot \nabla c-D\nabla^{2}c=0$$
where *c* and *D* = 10^−10^ m^2^/s denote the species concentration and the diffusion coefficient, respectively. Based on the above thesis, a three-dimensional simulation model was established using COMSOL Multiphysics (ver. 5.6) as provided in Appendix A.

## 3. Results and Discussion for Two-Fluid Micromixing

### 3.1. Effect of Channel Dimensions on ICEO-Based Microvortex Flow Behavior

As the microvortex flow behavior significantly affects the model performance, the generation and development of ICEO-induced microvortices above an SSFE are firstly investigated. Figure 3 presents the ICEO-microvortex variations with channel height *H* in the case of fixed electrode dimensions under a peak voltage (*A*) of 10 V and a frequency (*f*) of 100 Hz. When two fluids are separately introduced into the mixing channel, they are stirred from one side to the other and twist with each other once reaching SSFE, as shown in Figure 3a. The flow patterns in different cross sections that are normal to the flow direction appear as two opposite microvortices (O_1_ and O_2_) developing with each other and encountering each other on the SSFE. A stagnant line concurrently forms, and its position on the SSFE varies with mixing length *L*. In addition, the morphology of the ICEO-based microvortex pair is constantly evolving with increasing *H*, describing the process of microvortices developing within a constrained space. Specifically, two opposite peaks of flow velocity appear in the vicinities of both electrode edges, and they are antisymmetric at the symmetric cross section (b-b), while their amplitudes are asymmetric and the maximum amplitude appears above the SSFE edge near the channel side (herein left side on a-a section and right side on c-c section). It is also evident that the closer to the SSFE surface, the stronger the transverse flow field. The surface-averaged velocities *v*_ave_ in these three sections are shown in Figure 3c. It is worth noting that *v*_ave_ approximately reaches its plateau when *H* increases to 60 μm, and then it decreases slightly. This phenomenon is because the development of microvortices is suppressed by a lower channel and the channel top cannot be extended within a relatively higher channel. Additionally, the narrower the gap between 3DDE and SSFE is, the stronger the flow field is, mainly due to a dramatic voltage decrease along the narrower gap.

To better understand the development process of ICEO microvortices, two dimensionless parameters *R*_m_ and *y*_NF_ are defined to quantify the variation of microvortices on the SSFE. *R*_m_ is the ratio of the microvortex extension scope over the total space of a cross section, while *y*_NF_ refers to the normalized position of the stagnant line within the microchannel. It can be inferred from Figure 3d that both microvortices grow with *H*, and their total range gradually approaches a stable value. The microvortex that is away from 3DDE broadens more significantly, and the microvortex pair exhibits a symmetric distribution when both gaps are equal in width. Beyond this, the normalized position (*y*_NF_) of the stagnant line shows a sine form (*y*_NF_ = 0.0035 + Asin(π(*L*_N_ + 0.243)/B)) on the SSFE along with the channel length (Figure 3e). The coefficient A decrease linearly with *H*, while the coefficient B almost remains constant. As a result, the amplitude of the sine curve decreases with *H*, indicating the stronger suppression of the upper wall on one microvortex which is adjacent to 3DDE. In conclusion, on-demand microvortices could be generated in the presented model, and this can indeed possess the capacity to perform multicomplex micromixing.

### 3.2. Relation between Mixing Performance and Microvortex Index

The sinusoidally shaped microvortex pairs in the flow direction are applied to perform multi-fluid micromixing. Two-fluid micromixing is first investigated in the presence of an AC electric field with *A* = 10 V and *f* = 100 Hz, as shown in Figure 4. Figure 4a,b depict the mixing processes of two fluids with distinct concentrations of 0 and 1 mol/m^3^ at *H* = 20 μm and 60 μm, respectively. It can be obviously seen that the fluid concentration with an initial concentration of 0 mol/m^3^ becomes higher and higher as the mixing length (*L*) increases, while the one with an initial concentration of 1 mol/m^3^ becomes lower and lower, and both concentrations approach 0.5 mol/m^3^ when they are equal in flow rate. The concentration cross-sections at four flow positions show that the concentration distribution becomes more and more uniform and the concentration gradient becomes lower and lower. This phenomenon proves an efficient mass transport when two fluids pass through the SSDE region. In addition, the concentration gradient contours demonstrate that the contact interface oscillates from one side to the other due to the action of sinusoidally shaped ICEO-microvortex pairs. It is also worth noting that the contours of color extension at concentration sections are smaller at *H* = 60 μm than at *H* = 20 μm. This phenomenon also indicates that the microvortex development and the resulting mixing would be suppressed at a relatively lower channel height.

The mixing performance is quantified by calculating the mixing index (*M*_I_), where 0 and 1 represent unmixed and completely mixed cases. *M*_I_ is calculated by taking the ratio of the standard deviation of concentration at each sampling point, *σ*, to the maximum standard deviation, *σ*_max_, according to the following equation [39]:(6)$$M_{I}=\left(1-\sqrt{\sigma^{2}/\sigma_{\max}^{2}}\right)$$
where $\sigma =\sqrt{1/N\sum {\left(c_{i}-c_{\infty }\right)}^{2}}$, *c*_i_ (from 0 to 1) denotes the value of the concentration fraction at sampling point *i* (*i* = 0…n), and *c*_∞_ represents the concentration fraction value for full mixing.

Figure 4c clearly demonstrates that the convective flux increases to a stable value with *H* and dominates the mass transport between two fluids compared to the diffusion flux, indicating the high efficiency of the sinusoidally shaped ICEO-microvortex pairs for mixing. Figure 4d depicts the variations of *M*_I_ with the mixing length (*L*) under different channel heights. It is observed that the longer *L* is, the higher *M*_I_ is for better mixing performance. *M*_I_ gradually approaches its plateau at *H* = 60 μm and decreases with a further increase in *H*. Specifically, the optimal mixing performance is achieved at *H* = 60 μm, where a smaller mixing length (*L* = 2.1 mm) and a higher mixing efficiency (*M*_I_ = 0.922) are presented. This phenomenon typically relies on the microvortex development within a confined microchannel and is in excellent agreement with the variations of surface-averaged velocity shown in Figure 3c. Consequently, the mixer design with *H* = 60 μm is selected to perform the subsequent analysis.

As the ICEO-based microvortices dominate the mixing behavior, the enstrophy density is defined as [40]
(7)$$q=\sum_{i}q_{i}=\frac{1}{2}\omega_{i}\omega_{i}$$
where *ω* is the vorticity and *q* is a scalar term that consists of three components, *q*_x_, *q*_y_, *q*_z_. This term on a cross section will provide more evidence on the vortex development within the provided physical model and how each component contributes to the mass transport.

According to the equation, the vorticity distribution at cross sections with distinct vertical heights is first explored as shown in Figure 5a. It is clear that the vorticity becomes stronger and stronger as the cross section is approaching the electrode surface. The narrower the gap between one 3DDE and the SSFE is, the more significant the behavior of the vorticity on its side is. Next, the enstrophy density and its three components at a cross section of *L* = 0.35 mm are made non-dimensional by using the maximum average enstrophy density, as shown in Figure 5b. As presented in this figure, two peaks for each enstrophy density appear in the vicinities of both electrode edges, while the values in other positions go to zero. More importantly, the values of enstrophy density in the *x*-axis direction *q*_x_ are almost equal to the amplitude of *q* and much higher than the other two components; that is, the vortices stretching in the *yz*-plane make the greatest contribution to disturbing the incoming fluids.

Further, in order to consider how the microvortices contribute to mixing, a vortex index (*Ω*_I_) involving the vorticity history of a flow is proposed as follows:(8)$$\Omega_{I}=\frac{\int \left|\omega_{x}\right|dV}{Q}\frac{D}{w_{0}}$$
where *ω*_x_ and *Q* represent the vorticity in the flow (*x*-axis) direction and the flow rate, respectively. In this expression, the vortex index considers not only the local flow vorticity but also the vorticity history that the mixing process develops. In addition, the vortex index is nondimensionalized by taking the ratio of the hydraulic diameter, *D*, to the channel width *w*_0_.

Figure 5c shows that the vortex index *Ω*_I_ increases linearly in terms of mixing length L. It can also be observed in a specified cross section that *Ω*_I_ gradually increases to a peak at *H* = 60 μm and then slightly decreases with a further increase in *H*. Interestingly, this trend is in good agreement with *M*_I_. In terms of the mixing index variations, a functional form of *M*_I_ dependence on *Ω*_I_ can be obtained by fitting all the data of Figure 4d and Figure 5c, as plotted in Figure 5d. The functional relation between *M*_I_ and *Ω*_I_ can be modeled as
(9)$${Μ}_{Ι}=1-\exp(-0.0451\Omega_{I})$$

Herein, it can be inferred that perfect mixing (*M*_I_ > 0.9) can be achieved with *Ω*_I_ exceeding 51. Accordingly, this model can be utilized to evaluate the performance of a vortex mixer without solving the coupled diffusion and momentum equations.

### 3.3. Effect of Applied Voltage on Mixing Performance

The applied voltage not only determines the electric intensity within the microchannel, but also influences the induced voltage (*A*_f_) and the rendering zeta potential (*ζ*) on the SSFE. In terms of these that are directly related to the transverse flow intensity, the effect of applied voltage *A* on mixing performance is shown in Figure 6. Figure 6a exhibits that the amplitudes of both induced voltage *A*_f_ and line-averaged zeta potential *ζ*_ave_ increase linearly with increasing *A*, proving an enhancement of the flow intensity according to Equation (2). The non-zero *ζ*_ave_ also indicates the asymmetrically induced microvortex flow above SSFE, while the zero *ζ*_ave_ demonstrates the flow symmetry at the positions (e.g., section b-b) with equal gaps to 3DDEs. Additionally, the distributions of zeta potential *ζ* on the SSFE surface are shown in Figure 6b. Notably, *ζ* also exhibits linear variation from negative to positive along the line segment, and the maximum amplitude appears on the side of smaller gap, indicating a higher density of the induced bipolar double-layer diffuse charge. Interestingly, the positions *y*_NE_ where *ζ* = 0 V form a sine curve along the length of SSFE. This stagnant curve for *ζ* shares the same tendency as the variation of flow stagnant position shown in Figure 3e, benefiting the prediction of the microvortex flow pattern without solving the Navier–Stokes and continuity equations for the velocity and pressure fields. 

The surface plots in Figure 6c show the concentration distributions under voltages of 0, 6, 12, and 16 V. A clear interface and slight mixing due to the molecular diffusion process occur when the electric power is off. Once a voltage is applied to the 3DDEs at a frequency of 100 Hz, efficient mixing mainly caused by the convective mass transport is obtained. Figure 6d also shows that both *M*_I_ and *Ω*_I_ increase and the mixing length for full mixing (*M*_I_ > 0.9) decreases with increased applied voltage. In other words, an appropriate increment in the applied voltage can benefit the miniaturization of the proposed physical model. In addition, the variations of *M*_I_ and *Ω*_I_ as a function of applied voltage further confirm the validity of the presented theoretical formula. That is, a higher voltage would give rise to stronger microvortices, strengthen interface disturbance, and enhance the mixing of the working fluids.

### 3.4. Effect of AC Frequency on Mixing Performance

In terms of Equation (1), the ICEO microvortex is a sensitively frequency-dependent flow. Figure 7a shows that |*ζ*_ave_| exhibits a dramatic increase at a relatively lower frequency and attains a peak value at *f* = 100 Hz around the RC charging frequency *f*_RC_ = *σ*(1 + *δ*)/2π*C*_d_(0.195*W*_F_) = 138 Hz. Once *f* exceeds this frequency, |*ζ*_ave_| gradually declines to a stable value close to zero, due to the charge relaxation phenomenon forming the double layer. As a result, both *M*_I_ and *Ω*_I_ reach their peaks at *f* = 100 Hz, as depicted in Figure 7b, demonstrating an excellent electric condition for ICEO-based mixing within this presented physical model. In addition, there is almost no effective mixing in a DC limit frequency (e.g., *f* = 1 Hz in Figure 7c). That is due to the screening of the bipolar free surface charge and the corresponding extremely weak ICEO flow. It is also noted from the cross-sectional views of concentration that the liquid interface disturbance becomes weaker and weaker at higher frequencies and thus leads to worse and worse mixing performance (e.g., *f* = 600 Hz).

## 4. Analysis of Three-Fluid Sequential Micromixing

According to the above analysis, the microvortex pattern and its induced mixing sequence can be altered by a flexible design of electrode positions in a microchannel. Herein, an RFE is deposited at the entrance of the channel, aiming to disturb the contact surface S_12_ and perform mixing between one side fluid (fluid 1) and the middle fluid (fluid 2) firstly, as shown in Figure 8a. The RFE is followed by an SSFE, and it is expected that the initially mixed fluid mixes with the other side fluid (fluid 3) in this region. 

Figure 8b depicts the mixing evolution of the middle fluid and both side fluids along the outflow direction. Two clear interfaces are visible at the entrance of the main channel (*L* = 0 mm), and then the contact surfaces S_12_ and S_23_ are gradually destroyed one after another when passing through the regions in which the RFE and SSFE are deposited. A relatively uniform concentration distribution (0.34~0.36) shown in the cross section at *L* = 3.1 mm demonstrates that the middle fluid can mix with both side fluids. It can be inferred from Figure 8c that two side fluids separated by the middle fluid can also mix with each other eventually. These phenomena have successfully proved the validity of the sequential mixing concept. 

To obtain an optimized sequential mixer in which fluid 1 and fluid 2 mix as much as possible while fluid 3 is involved as little as possible in the first mixing stage, the variations of the microvortex pattern on RFE with respect to *W*_F1_ are investigated, as shown in Figure 8d,e. As shown in Figure 8d, the magnitude of *ζ* along the electrode width increases with increasing *W*_F1_, and the stagnant line where *ζ* = 0 V gradually migrates to the channel center. As a consequence, there are the regions in which both microvortices linearly broaden as a function of *W*_F1_, as illustrated in Figure 8e. The total extension for O_1_ + O_2_ is about 0.51 of the channel width at *W*_F1_ = 40 μm, indicating inadequate participation of middle fluid in the first mixing stage. However, it increases nearly to 0.8 of the channel width at *W*_F1_ = 100 μm, giving rise to an undesired disturbance of the other side fluid. It is worth noting that the effective action scope of microvortices O_1_ + O_2_ can extend to 0.66 of the channel at *W*_F1_ = 70 μm, demonstrating that only the middle fluid and one of the side fluids are involved.

To further examine and certify the sequential mixing concept, the mixing of middle fluid to side fluids one by one is conducted, as shown in Figure 9. The mixing behaviors in three RFE dimensions *W*_F1_ = 40, 70, and 100 μm are presented in Figure 9a, Figure 9b and Figure 9c, respectively. The concentration distributions in the XY-plane clearly show the evolution of sequential mixing. It can be observed that the middle fluid is not completely involved in first-stage mixing from a cross section (*L* = 0.3 mm) above a narrower RFE (e.g., *W*_F1_ = 40 μm); the contact surface S_23_ is almost undisturbed (*L* = 0.3 mm) and exhibits uniform mixing (*M*_I_ = 0.623) for fluid 1 and fluid 2 from a cross-sectional concentration gradient (*L* = 1.1 mm) with an appropriate RFE (*W*_F1_ = 70 μm), while S_23_ is perturbed and three-fluid mixing occurs above a relatively wider RFE (*W*_F1_ = 100 μm). As a result, the mixing performance in the first-stage mixing (1′-1′) is enhanced with an increase in *W*_F1_, as shown in Figure 9d. The perfect performance with *M*_I_ > 0.95 at the end of the channel demonstrates that the middle fluid can fully mix with both side fluids with the presented electrode configuration. Even if there is incomplete mixing above the RFE, the second mixing stage on the SSFE could still enhance the mixing. These phenomena are in excellent agreement with the above prediction using the ICEO-based microvortex pattern. 

Figure 9e shows that the inlet flow rate also affects the mixing behavior significantly. Both mixing indexes for the first and second mixing stages decrease with the increase in the inlet flow rate, mainly due to the increased contact-interface strength and the reduction in residence time along the electrodes. When the inlet flow rate exceeds 2 mm/s, the mixing index decreases to lower than 0.58, indicating an incomplete mixing between fluid 1 and fluid 2 in the first stage. Consequently, a higher voltage or longer electrode design is needed to achieve desired sequential mixing at increased flow rates.

## 5. Conclusions

In this work, sequential micromixing between parallel streams utilizing ICEO-based microvortices is developed for the first time. An SSFE is utilized to generate time-varying microvortex pairs in the flow direction to disturb the contact interfaces between different streams. The development of the microvortices on the SSFE and the corresponding mixing as a function of channel height are investigated. It is found that the microvortex is suppressed within a relatively lower channel and could not extend to the entire space within a higher channel. Considering the flow history and the vortex intensity, a normalized mathematical model, namely the vortex index, is proposed to predict the mixing performance. The mixing performance shows a negative exponential distribution with respect to the vortex index, and it achieves a perfect behavior (*M*_I_ > 0.9) when the vortex index is higher than 51. In addition, a relatively higher voltage or lower flow rate contributes to a short mixing length. Specifically, the mixing efficiency reaches 0.920 even at a mixing length of 1.1 mm where the vortex index reaches 62.9 at *A* = 16 V and *v*_0_ = 2 mm/s. For three-fluid sequential mixing, an RFE is asymmetrically arranged at the entrance of the main channel to enhance the mixing sequence. In this electrode configuration, two streams flowing through the RFE perform the first mixing behavior, followed by the second mixing for the mixed stream and the other side stream. In particular, in the cases of *A* = 10 V and *f* = 200 Hz, a sufficient mixing with a mixing index of 0.623 between one side stream and the middle stream is achieved in the first mixing stage when the RFE is designed to be 70 μm wide at *H* = 60 μm, and a perfect entire-fluid mixing with a mixing index of 0.983 in second mixing stage is subsequently obtained. Consequently, the presented mixing model provides a promising approach for the design of a chaotic mixer, and the sequential mixing strategy can further expand its utility in biological and chemical assays.

## Figures and Tables

**Figure 1 micromachines-13-01985-f001:**
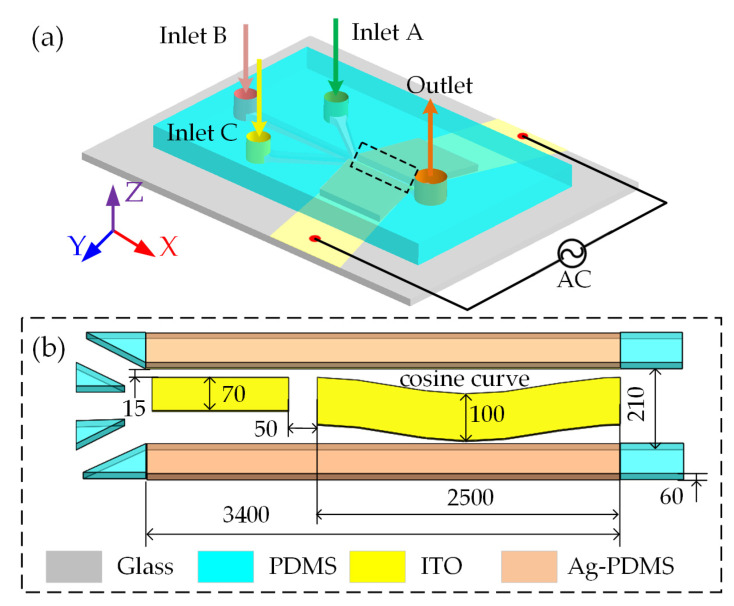
Schematic illustration of the electrokinetic micromixer: (**a**) three-dimensional model of the micromixer; (**b**) specific dimension (in units of microns) of the micromixer.

**Figure 2 micromachines-13-01985-f002:**
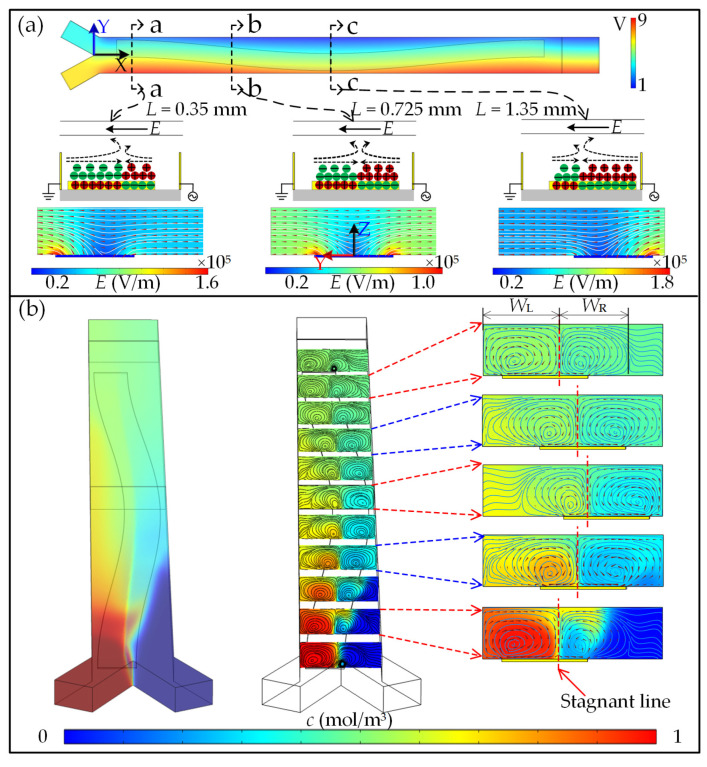
Mixing mechanism with asymmetric ICEO flow generated above a sinusoidally shaped floating electrode (SSFE): (**a**) electric field distribution at different sections in flow direction; (**b**) microvortex reconfiguration and the corresponding micromixing in the main channel.

**Figure 3 micromachines-13-01985-f003:**
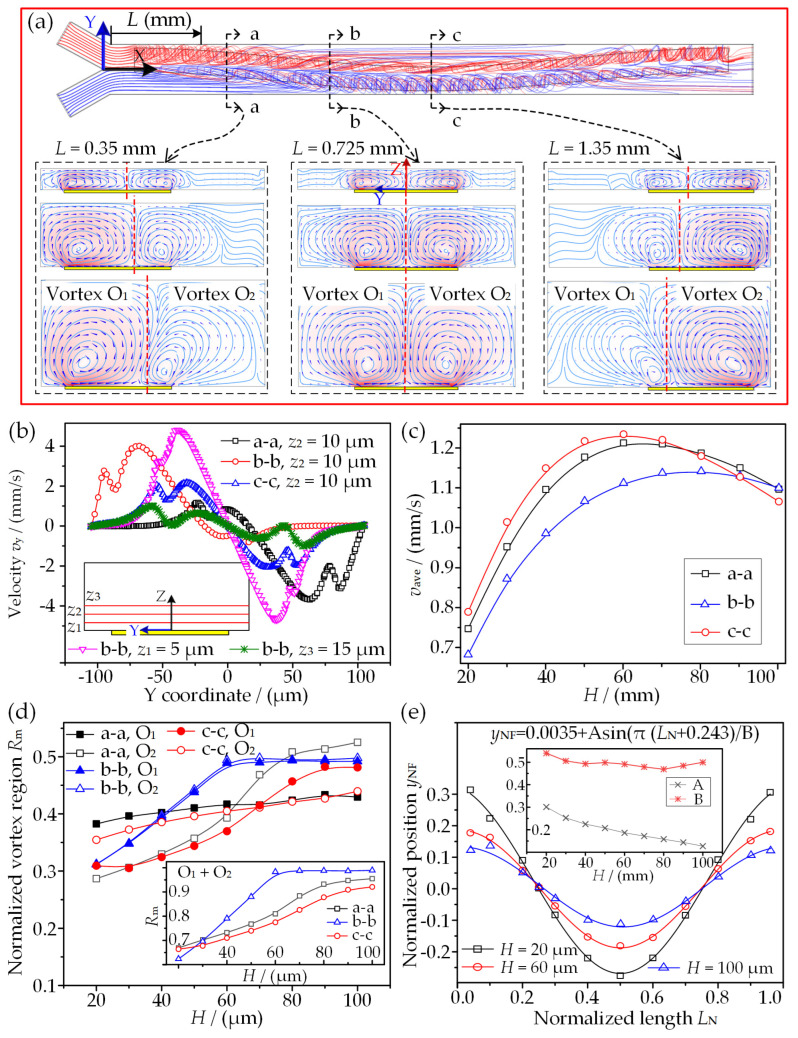
ICEO-microvortex analysis on the SSFE: (**a**) microvortex pattern varying with channel height *H* at different sections (a-a, b-b, and c-c); (**b**) velocity profiles along with the channel width; (**c**) averaged velocity on the selected cross sections in response to *H*; (**d**) the microvortex pair variations with *H*; (**e**) variations of the flow stagnant position above SSFE.

**Figure 4 micromachines-13-01985-f004:**
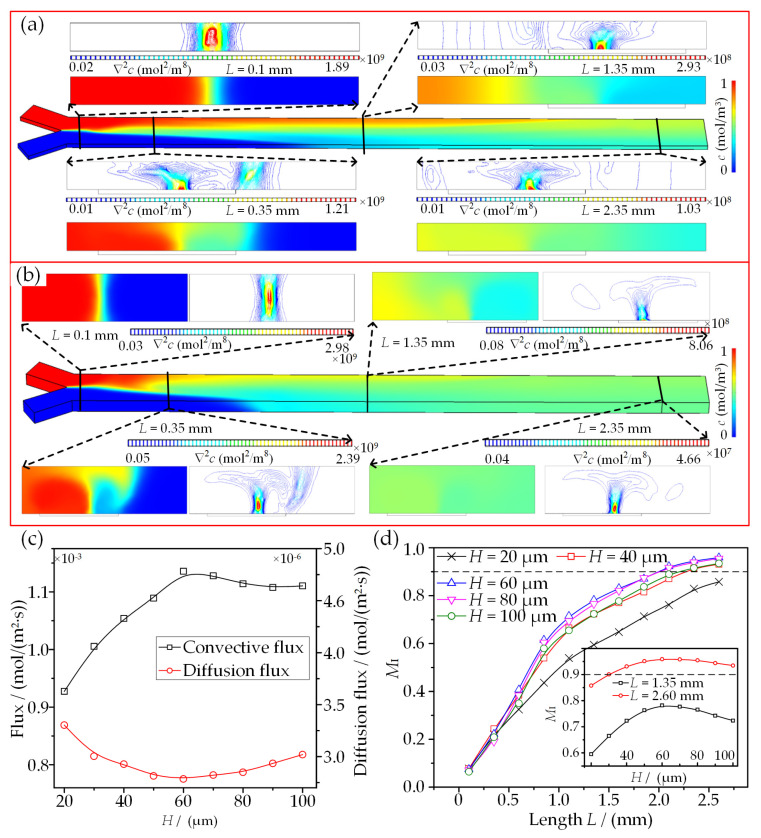
Comparison of mixing performance among different channel heights at *A* = 10 V and *f* = 100 Hz. (**a**,**b**) Concentration distribution and the corresponding concentration gradient at *H* = 20 μm and 60 μm. (**c**) Flux variations between two fluids. (**d**) Mixing efficiency vs. the mixing length under different channel heights *H*.

**Figure 5 micromachines-13-01985-f005:**
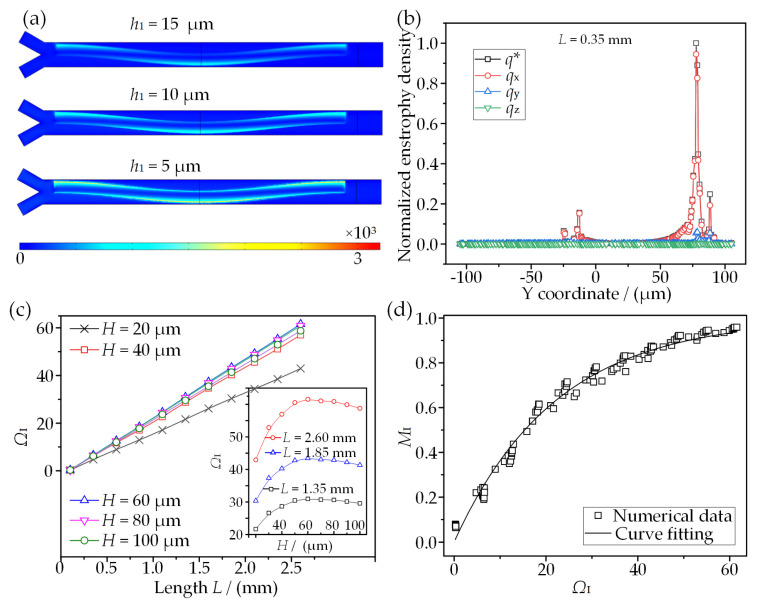
Analysis of the relation between vorticity and mixing performance: (**a**) distributions of vorticity magnitude in three cross sections; (**b**) profiles of mean enstrophy density and its three components; (**c**) variations of *Ω*I along the mixing length under different H values; (**d**) variations of MI with respect to *Ω*_I_.

**Figure 6 micromachines-13-01985-f006:**
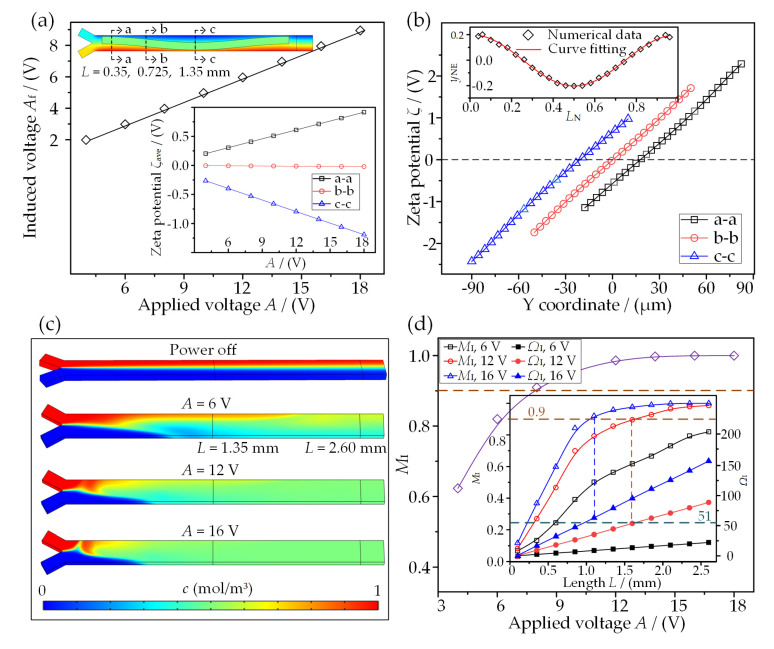
Influence of applied voltage *A* on mixing performance: (**a**) the induced voltage and the corresponding zeta potential on the SSFE; (**b**) variations of zeta potential on the SSFE; (**c**) concentration distributions under different voltages; (**d**) the mixing index as a function of *A*.

**Figure 7 micromachines-13-01985-f007:**
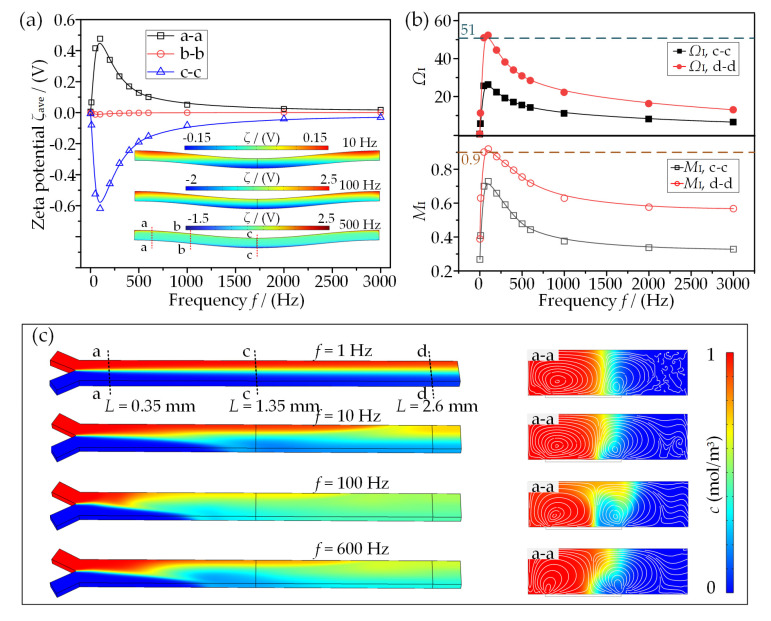
Influence of AC frequency *f* on the mixing performance: (**a**) variations of averaged zeta potential with *f* on three selected lines; (**b**) the mixing index and vortex index in terms of *f*; (**c**) concentration distributions under different AC frequencies.

**Figure 8 micromachines-13-01985-f008:**
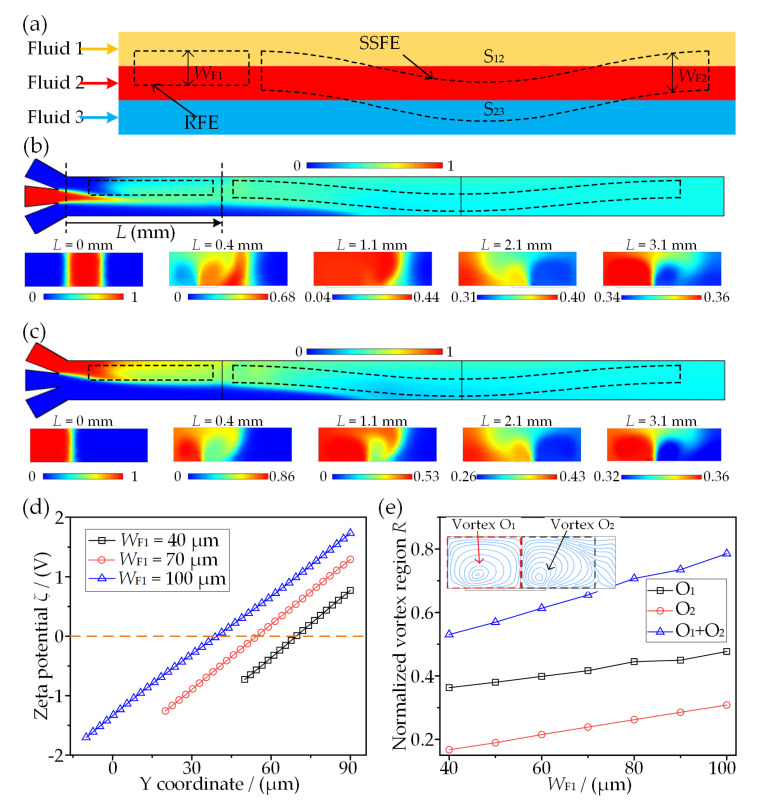
Influence of the rectangle floating electrode (RFE) on the performance of sequential micromixer. (**a**) Configuration of RFE and SSFE for sequential multi-fluid mixing. (**b**) Evolution of concentration distribution for middle fluid. (**c**) Evolution of concentration distribution for side fluids. (**d**) Variations of zeta potential on RFE. (**e**) The microvortex variation above RFE with different values of electrode width *W*_F1_.

**Figure 9 micromachines-13-01985-f009:**
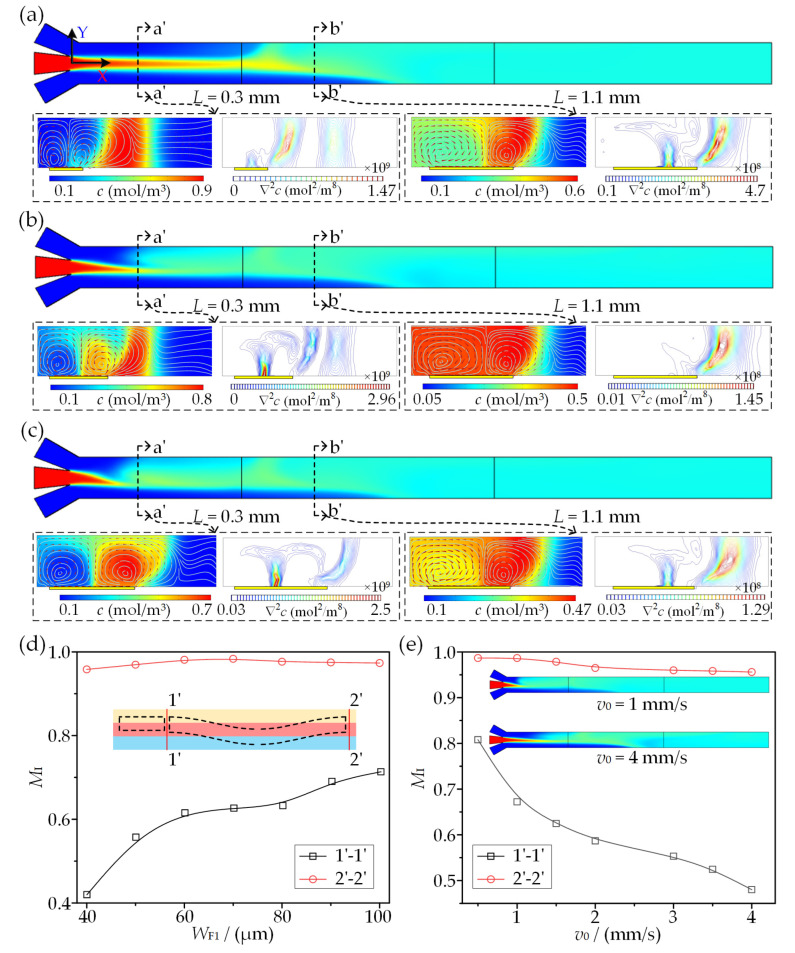
Analysis of mixing performance for three-fluid sequential micromixing at *A* = 10 V and *f* = 200 Hz. (**a**–**c**) Concentration distribution and the corresponding concentration gradient at *W*_F1_ = 40, 70, and 100 μm, respectively. (**d**) Mixing efficiency at cross sections after SFE and SSFE in terms of *W*_F1_. (**e**) Influence of inlet flow velocity on mixing index.

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
