# Peer review of "Analysis of Sequential Micromixing Driven by Sinusoidally Shaped Induced-Charge Electroosmotic Flow"

_micromachines, 2022, doi:10.3390/mi13111985_

Round 1

Reviewer 1 Report

This manuscript introduces a novel strategy for sequential micromixing between multiple parallel fluids utilizing Induced-Charge Electroosmotic flow. This manuscript is well organized and would be of interest to the readers of Micromachines. This reviewer is pleased to recommend the manuscript for publication after the authors address the following comments:

1. The micromixer consists a pair of 3D electrodes and a deposited electrode, but their heights and length are not presented, please provide the detailed dimensions.

2. It seems that the conductivities of parallel streams are not mentioned in the manuscript, please give the related information.

3. In Figure 3e, it is better to specify the profiles of parameters A and B, which may be helpful to demonstrate the variations of flow stagnant position.

4. The mixing length L should be defined in Figure 3a to help readers easily catch the key points.

5. In the simulation, the authors only provided the geometric model. More details, such as computational domains, initial conditions, and so forth, needs to be added in the manuscript.

Reviewer 2 Report

This research developed the multiple parallel stream mixer using the combination of RFE and SSFE technology.

The introduction part was a compact and reasonable demonstration of research needs and their goal.

The research design and results were clearly demonstrated with mathematical approaches and experimental results.

In my opinion, additional applications, such as chemical reagent mixing, or biological reagent mixing, might be presented for a better research paper.

I accept this paper in Micromachines.
